# Anti-VEGF Treatment of Diabetic Macular Edema in Denmark: Incidence, Burden of Therapy, and Forecasting Analyses

**DOI:** 10.3390/jpm13030546

**Published:** 2023-03-18

**Authors:** Yousif Subhi, Ivan Potapenko, Javad Nouri Hajari, Morten la Cour

**Affiliations:** 1Department of Clinical Research, University of Southern Denmark, DK-5230 Odense, Denmark; 2Department of Ophthalmology, Rigshospitalet Copenhagen, DK-2600 Glostrup, Denmark; 3Department of Clinical Medicine, University of Copenhagen, DK-2200 Copenhagen, Denmark

**Keywords:** diabetic macular edema, anti-VEGF, incidence, burden of therapy, forecasting, Denmark

## Abstract

Background: The aim of this study was to analyze demographically stratified incidence rates of patients with diabetic macular edema (DME) commenced in anti-VEGF therapy, to study temporal trends, to report the proportion of patients in active therapy over time, and to develop a model to forecast the future number of patients in active treatment. Methods: This was a retrospective registry-based study of all patients with DME who received at least one intravitreal anti-VEGF treatment from 1 January 2007 to 30 June 2022. Population data were extracted from Statistics Denmark. Results: This study included 2220 patients with DME who were commenced in anti-VEGF therapy. Demographic analyses revealed higher incidence rates among males than females and among those aged 60–80 years. The number of patients in active treatment followed an exponential decay curve; hence, this was used to mathematically model the number of patients in active therapy. The number of patients in active treatment is expected to stay relatively stable with a minimal increase until the year 2023. Conclusions: This study provides insight into the practical aspects of the anti-VEGF treatment of DME that allow the planning of adequate health services.

## 1. Introduction

Diabetes is one of the most prevalent diseases with global estimates of 533.6 million individuals with diabetes [1]. This number is expected to increase, albeit with variating speed in different countries [1]. Diabetes leads to a range of complications throughout the body, including diabetic eye disease. Diabetic retinopathy and maculopathy are the cornerstones of diabetic eye disease and together brings diabetic eye disease to one of the leading causes of vision loss worldwide [2]. The retinopathy in diabetes is a consequence of a complex interplay between retinal ischemia, vascular changes, dysfunction of the inner blood–retina barrier, and expression of vascular endothelial growth factor (VEGF) [3,4,5]. These pathophysiological circumstances can eventually lead to a clinically observable macular edema [3,4,5]. Since diabetic maculopathy, or the more descriptive term diabetic macular edema (DME), predominantly affects the macula, clinically significant cases can impair vision and need treatment. Hence, DME is an important cause of visual impairment from diabetic eye disease.

Treatment of DME can save or improve vision. Macular laser treatment for DME was practiced for many years in which focal laser photocoagulation was applied to reduce macular edema, halt the worsening of vision, and in some cases improve visual acuity [6]. The exact mechanism by which macular laser was effective remains to be fully elucidated [5], but its efficacy was superior to that of observation [5,6]. After the introduction of anti-VEGF therapy, pivotal large multicenter randomized controlled trials determined the superior efficacy of anti-VEGF against DME, and anti-VEGF treatment as first-line therapy has since been the cornerstone of DME management [7,8,9,10].

Intravitreal anti-VEGF therapy demands repeated injections over time. From a healthcare planning perspective, it is important to understand the number of injections over time in relation to patient burden to better plan adequate treatment facilities and services [11,12]. Unlike neovascular age-related macular degeneration (AMD), which requires continuing treatment for years for the majority of patients [13,14,15,16], in DME, studies report that the burden of anti-VEGF treatment is mainly in the first years of therapy [17,18,19,20,21]. This circumstance for DME presents an entirely different consideration than, e.g., neovascular AMD, in terms of the number of patients in active therapy when planning a large-scale, anti-VEGF therapy service for DME.

In this study, we extracted a large dataset from one of Europe’s largest departments of ophthalmology and extracted corresponding population census data to better understand the incidence of patients with DME commenced in anti-VEGF therapy and to understand temporal trends and burden of therapy as the proportion of patients in active therapy over time and developed a model to forecast the future number of patients in active treatment.

## 2. Materials and Methods

### 2.1. Study Design

This was a retrospective study of patients with DME in the Capital Region of Denmark (~1.9 million inhabitants). The Capital Region of Denmark is the most populous region of Denmark and includes the Danish capital Copenhagen. Patients included were those with DME who received anti-VEGF therapy from 1 January 2007 to 30 June 2022. This was a registry-based study. Study approval was obtained from the department, from the Danish Data Protection Agency (jr. no. FSEID-00006126), and from the Centre of Regional Development/Institutional Review Board (jr. no. R-20052134). All aspects of this study adhered to the tenets of the Declaration of Helsinki.

Study data were extracted from the clinical database BOB that stores data on all patients in anti-VEGF treatment in the Capital Region of Denmark [22], including information on the retinal diagnosis and time of administration of anti-VEGF treatment. All population census data were extracted from Statistics Denmark [23], which is the statistical authority in Denmark. Statistics Denmark compiles data on population count, immigration and emigration trends, live births, and deaths. These data are provided to the Danish Institute for Economic Modelling and Forecasting (DREAM)—an independent, semi-governmental institution that provides forecasting analyses of the Danish society and economy [24]. Statistics Denmark then publishes the most likely model of the future forecast of the Danish population based on DREAM analyses [24].

### 2.2. Clinical Pathway of Patients with Diabetic Macular Edema

Public tax-based healthcare coverage allows all Danish citizens the right to access health services, free of charge. Access to specialized health care is managed through general practitioners. One of the few examples is the access to specialized eye care. Patients can book an appointment with a private ophthalmologist without the need for a referral from their general practitioner or other health care professional. For patients with diabetes, nationwide programs promote systematical examination of patients including relevant biochemistry at regular intervals, and patients are also seen by a range of specialists to screen for and treat complications, including regular retinal examinations. According to the Danish national guidelines [25], patients with diabetes are screened according to an individualized screening interval (Table 1). Screening of diabetic retinopathy and maculopathy is based on fundus photography and optical coherence tomography (OCT) upon suspicion of maculopathy.

In Denmark, only selective public hospital departments, and no private hospitals, can provide on-label anti-VEGF treatments as outlined by the national health law. A few clinics provide off-label anti-VEGF (i.e., bevacizumab) treatments for a few patients for various reasons. Thus, almost all patients in need of anti-VEGF treatment are referred to one of these public hospital departments for evaluation and commencement of anti-VEGF treatment. In the Capital Region of Denmark, all referrals are made to the Department of Ophthalmology, Rigshospitalet, and thus registered in the BOB database if the retinal specialist commences anti-VEGF treatment. This unique organization in a single center with high expertise allows a combination of high compliance to ophthalmic therapy in patients who are otherwise low compliance to therapy and the possibility to conduct large-scale epidemiological studies of disease epidemiology and treatment burden [12].

### 2.3. Treatment Commencement, Regimen, and Discontinuation

All patients had best-corrected visual acuity (BCVA) measured with Snellen chart, slit-lamp biomicroscopy, dilated retinal examination, and macular OCT. Macular OCT was made using Topcon Triton swept-source OCT (Topcon Corporation, Tokyo, Japan) or Heidelberg Spectralis spectral-domain OCT (Heidelberg Engineering, Heidelberg, Germany). Fluorescein angiography was performed in select cases of uncertainty on the source of macular edema. Treatment was commenced according to the local department guidelines, which indicates that anti-VEGF treatment is commenced with 1–3 injections depending on the severity of the DME. Patients are re-evaluated 1–2 months after their last anti-VEGF injection, and re-treatment is commenced depending on the presence of macular edema. If the macular edema is resolved, the patient is seen at 2–4 months intervals at the physician’s discretion. According to national guidelines, Ranibizumab (Novartis, Basel, Switzerland) was the first choice of treatment until 2013, and Aflibercept (Bayer, Leverkusen, Germany) was the first choice of treatment from 2014 and onwards. Patients could be changed from Aflibercept to Ranibizumab and vice-versa upon lack of treatment response evaluated at the physician’s discretion. Lack of treatment response on anti-VEGF would lead to discontinuation of anti-VEGF therapy and initiation of local photocoagulation therapy and/or intravitreal dexamethasone therapy at the physician’s discretion.

For proliferative diabetic retinopathy, panretinal photocoagulation was the first choice, and cases could be commenced in anti-VEGF therapy or vitrectomy on a case-by-case basis, where panretinal photocoagulation treatment was insufficient in obtaining disease control. Patients with proliferative diabetic retinopathy treated with anti-VEGF therapy without the presence of any DME were not included in this study.

Upon complete resolution of anti-VEGF treatment without the need for treatment for 6–12 months, cases were discontinued for follow-ups at national retinal screening sites or at private ophthalmologists every 3–4 months for the rest of their life. Treatment was also discontinued in cases with a lack of treatment response despite anti-VEGF or other treatments and where BCVA ≤ 0.05 Snellen.

### 2.4. Data Analysis and Statistics

Patients were included in our data if they had received at least one anti-VEGF injection. All patients were only included once, regardless of one or both eyes were treated. All statistical analyses were made in Python v. 3.11. For curve fitting, we employed the Levenberg–Marquardt algorithm in the SciPy package v.1.10.0. The incidence of patients with DME in anti-VEGF treatment was calculated as the ratio between the number of patients receiving their first anti-VEGF injection (in either eye) and the population number in the Capital Region of Denmark. This incidence was also calculated using the appropriate age- and sex-stratified patient data and census data. The number of patients in active therapy followed an exponential decay curve irrespective of the time of treatment commencement. Thus, we used this observation to mathematically express the numbers as *a∙e^−k·t^* in which *a* and *k* are parameters fitted using historical data. Based on this, a mathematical model was created to express the number of actively treated patients by only using the two fitted parameters and the number of new patients in any year:(1)$$A\left(T\right)=\overset{T}{\underset{t=1}{\sum }}n\left(t\right)\cdot e^{-k\cdot \left(T-t\right)}$$
which allowed us to express the number of actively treated patients (*A*) by only using the two fitted parameters and the number of new patients (*n*) in any year (*T*).

This model allows prediction of the future number of patients when also using data on predicted population data and age-stratified incidence of patients with DME in anti-VEGF treatment. This model assumes neither temporal changes in the indication nor efficacy of the given treatment. This mathematical approach is previously described in detail by Potapenko and la Cour [12].

## 3. Results

### 3.1. Number of Patients and Temporal Trends

During the study period, 2220 patients with DME were commenced in anti-VEGF therapy and 1953 patients were later discontinued from anti-VEGF therapy. We saw a gradual increase in the number of patients commenced in anti-VEGF treatment until the year 2019. Discontinuation of patients from anti-VEGF therapy also increased gradually. From the year 2020 and onwards, due to the COVID-19 epidemics and its impact on referral and discontinuation patterns, we observed a shift towards a lower number of new patients and an increasing discontinuation. Temporal trends of patients with DME in anti-VEGF therapy are illustrated in Figure 1.

### 3.2. Population Incidence of Patients with Diabetic Macular Edema Commenced in Anti-VEGF Therapy

The annual incidence of DME commenced in the Capital Region of Denmark in 2012–2021 was on average 193 individuals, which is equivalent to 10.7 per 100.000 inhabitants annually. Age- and sex-stratified incidence rates are presented in Table 2. The incidence of DME was higher in males than females and peaked among individuals aged 60–80 years.

Using age- and sex-stratified population forecasts for the number of individuals in the Capital Region of Denmark, we extrapolated the number of patients with DME commenced in anti-VEGF therapy until the year 2035 (Figure 2). Based on these analyses, we expect an annual incidence rate of ~160–175 individuals with a minimal increase over time.

### 3.3. Proportion in Active Anti-VEGF Treatment over Time

The proportion of patients with DME in active anti-VEGF treatment was 61.1% after year 1, 48.2% after year 2, 36.9% after year 3, 31.2% after year 4, 25.1% after year 5, and 10.2% after year 8 (Figure 3). These observations substantiate two important messages. First, approximately half of all patients remain in anti-VEGF therapy for 2 years. Second, the number of patients in active treatment exhibited a pattern of decrease relative to its previous value, i.e., an exponential decay. An exponential decay can be mathematically expressed using f(t) = *a∙e^−k·t^*, which in this case can be modelled to *a* = 0.601, *k* = 0.217, and *t* = time from commencement of anti-VEGF therapy (Figure 3). The model had a high goodness of fit at *R*^2^ = 0.81. According to our model, we expect that only 60.1% will remain in therapy after the first year and that the number of patients in active treatment will decrease by 21.7% for each following year.

### 3.4. Modelling the Future Number of Patients with Diabetic Macular Edema in Anti-VEGF Treatment

Based on the historical data, our model and population forecasts, we extrapolated the number of patients in active anti-VEGF therapy (Figure 4). We estimate stability in the number of patients in active anti-VEGF therapy with very small expected growth, at least until the year 2035. Our analyses exclude historical data for 2020 and 2021 from the model due to the COVID-19 epidemics and its impact on referral and discontinuation patterns; thus, these numbers assume a return to pre-COVID-19 state of normality.

## 4. Discussion

This study allows insight into important aspects of the trends in anti-VEGF therapy of DME in the Capital Region of Denmark. In the years 2007–2011, we observed an adaptation trend with an increasing number of patients referred for therapy. The following years had a stable number of new patients. The discontinuation trend exhibited an almost delayed pattern in comparison to that of new patients, which reflects the fact that a large number of patients were discontinued from therapy after a few years. The increase in discontinuation of therapy in 2021–2022 should be interpreted with caution, since this period is also affected by various circumstances related to the COVID-19 epidemics. Analysis of patient demographics showed that the incidence of DME commenced in anti-VEGF therapy was higher in males than females and that the incidence peaked in the age range of 60–80 years. These findings are in line with epidemiological reports of DME [26].

For patients in active treatment, we observed that in a period of stability (i.e., after the adaptation stage and prior to the COVID-19 epidemics), there is an interesting phenomenon of stability in the number of patients with DME in active therapy while the patient population itself is subject to a continuous change. This pattern is observed for the years 2015–2019 in Figure 2 and Figure 3, as well as the forecasts in Figure 3. When planning an anti-VEGF therapy for patients with DME, our observations indicate that efforts need not be made into expanding the future number of injection rooms or personnel, but instead on patient education and circumstances around the intake of new patients and efforts regarding discontinuation, which include information regarding the condition from a tertiary specialized healthcare organization to a primary healthcare unit for future screening and observation. The importance of this cannot be overstated; in patients with retinal diseases, those with DME rank poorest in health literacy [27].

The DRCR Retina Network Protocol T Extension Study reported treatment and clinical outcomes at 5 years after the commencement of anti-VEGF therapy [18]. This was an extension study to the original Protocol T study, which randomized and compared the efficacy of aflibercept, bevacizumab, or ranibizumab [17]. The extension study of 317 patients with DME found that after the first 2 years, only a minority were in need of further anti-VEGF injections [18]. Large real-life studies with long-term outcome data (≥5 years) of treatment burden are limited. The VISION study reported real-world treatment patterns for eyes with DME treated with anti-VEGF for at least 3 years [19]. This multicenter study of nine Belgian clinics reported long-term efficacy for 55 patients with DME [19]. Here, the median number of injections for all patients was 5.0 in the first year, which fell to 3.0, 1.0, 0.0, and 0.0 in the 2nd, 3rd, 4th, and 5th years, respectively [19]. Zirpel et al. reported Swiss long-term outcome data for 191 patients with DME and found that number of injections fell dramatically after the first year, and that loss to long-term follow-up was mainly due to discontinuation and referral back to the private ophthalmologist for screening and observation [20]. A multicenter retrospective study of 12 institutions in Latin America and Spain included 201 patients with DME for a study of 5-year outcomes after anti-VEGF therapy [21]. This study reported a gradual decline in the mean number of injections per year at 3.3, 2.1, 1.5, 1.3, and 1.2 for 1st, 2nd, 3rd, 4th, and 5th years, respectively [21], which from our point of view also exhibits an exponential decay pattern. Interestingly, many such studies report the number of injections for the entire population, but not the number of patients in active treatment, which only gives partial insight into the actual burden of treatment. In our study of 2220 patients with DME with follow-up data for up to 14 years, we are able to confirm the notions of other real-life studies on a larger scale.

The limitations of this study should be kept in mind. First, this was a retrospective database study of routine patients with DME commenced in anti-VEGF therapy, and therefore, data did not include patients with DME who did not have CSME, patients with DME who did not want anti-VEGF therapy for any reason, or were ineligible for anti-VEGF therapy for any reason. This means that our incidence data will underestimate the total number of patients with DME. Second, some of the analyses exclude data from 2020 and 2021 due to the COVID-19 epidemics, and we forecast future numbers under the assumption that we at some point will return to a pre-COVID-19 state of normality. Only time can tell if that is true. Third, our population forecasts rely on the accuracy of calculation from Statistics Denmark, which is based on a range of assumptions, which are best guesses—not facts. Fourth, the recent approval of brolucizumab and faricimab has introduced new anti-VEGF drugs for which long-term real-life data for DME do not exist. Thus, there is also the uncertainty of whether the pattern of decay of those in treatment will remain the same. Finally, the incidence of diabetic retinopathy or DME is correlated to blood glucose control. In the Capital Region of Denmark, the Steno Diabetes Center organization, which is a specialized diabetes hospital that works as an integrated part of the public healthcare system, has been acknowledged for its contribution to improved blood glucose control and a low rate of microvascular complications [28,29,30]. Populations with different levels of blood glucose control may experience different incidence rates of DME. A difference in blood glucose control may also play a role in the long-term need for anti-VEGF therapy. These circumstances should be considered as a limitation in the generalizability of our results to other populations.

In conclusion, we here report the incidence of patients with DME commenced in anti-VEGF therapy and present temporal trends. We confirm the findings of previous studies that the burden of treatment is mostly within the first couple of years. Our analyses reveal that the number of patients in active anti-VEGF therapy remains at a more or less constant number, while the individual patients change. These results and our forecasting model provide key insight into the planning of health services and highlight specific points for consideration. Importantly, when designing and planning anti-VEGF treatments for patients with DME, upon reaching a steady state in a Northern European demographic, the number of patients in anti-VEGF therapy is expected to remain more or less stable with no significant change, at least until the year 2035.

## Figures and Tables

**Figure 1 jpm-13-00546-f001:**
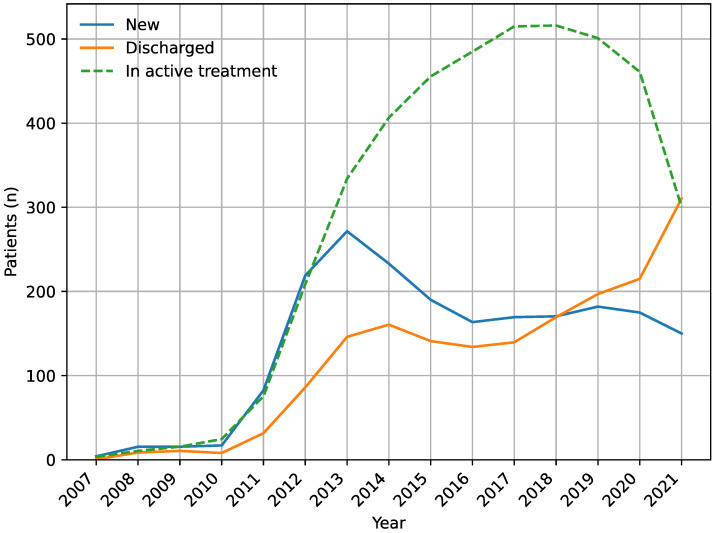
Temporal trends in patients with diabetic macular edema commenced in anti-VEGF therapy (blue), patients discontinued from anti-VEGF therapy (orange), and patients in active anti-VEGF therapy (green). Of note is the introduction of anti-VEGF therapy in 2007 for age-related macular degeneration and its gradual use for diabetic macular edema, and the impact of COVID-19 epidemics and on referral and discontinuation patterns from 2020 and onwards.

**Figure 2 jpm-13-00546-f002:**
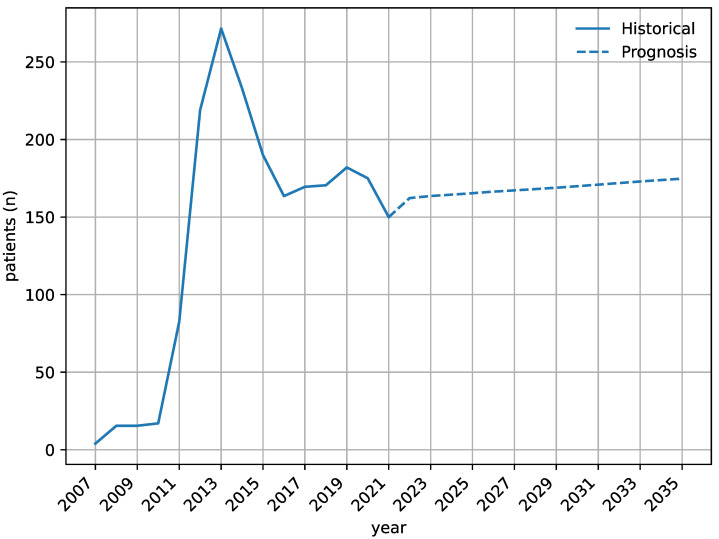
Forecasting of future patients with diabetic macular edema using age- and sex-stratified incidence rates combined with the prognostic expectations of Statistics Denmark until the year 2035.

**Figure 3 jpm-13-00546-f003:**
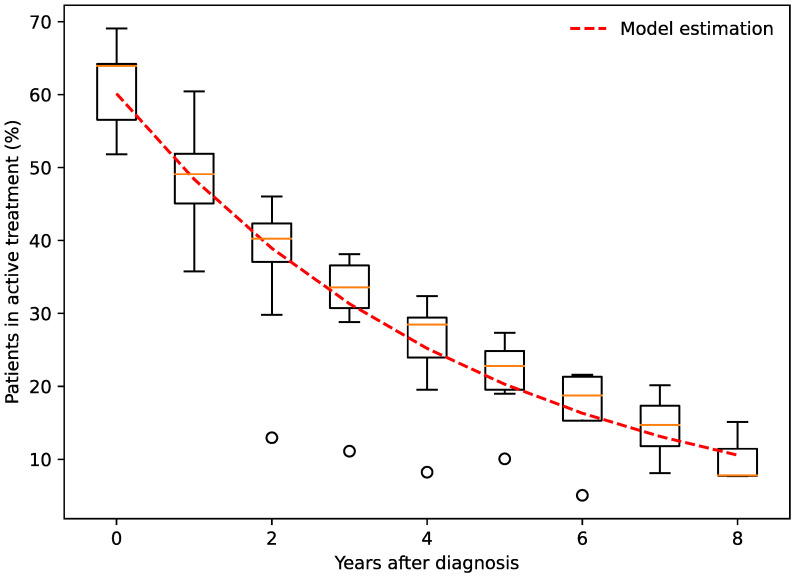
Patients with diabetic macular edema in active anti-VEGF treatment in years after the initial time of diagnosis and commencement of anti-VEGF therapy. Orange line indicates the percentage of patients in active treatment, box and whiskers indicate the interquartile range and 95% confidence interval. We calculated an exponential decay model fit (in red) with a high goodness of fit (*R*^2^ = 0.81) for *f*(*t*) *= a∙e ^kt^*, where *a* = 0.601, *k* = 0.217, and *t* = time from diagnosis.

**Figure 4 jpm-13-00546-f004:**
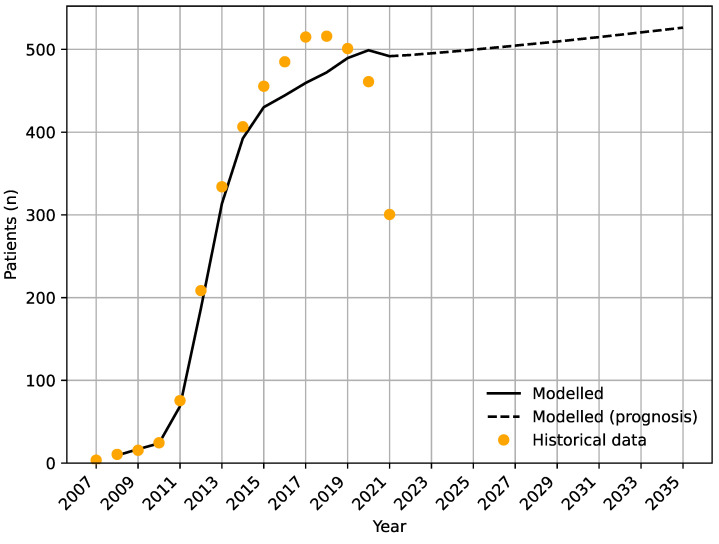
Prognosis of patients with diabetic macular edema in active anti-VEGF therapy as estimated by the expected future new cases and our model of the discontinuation rate. Forecasts are made until the year 2035.

**Table 1 jpm-13-00546-t001:** Screening intervals for patients with diabetes according to the Danish national guidelines.

Level of DR	Subgroup	Diabetes in Control *	Diabetes Not in Control or No Information
0: No DR		24–48 ** months	12–24 months
1: Mild NPDR	No DME	24 months	12 months
	DME without CSME	3–6 months	3 months
2: Moderate NPDR	No DME	12–24 months	6–12 months
	DME without CSME	3–6 months	3 months
3: Severe NPDR	No DME	3–6 months	3 months
	DME without CSME	3 months	3 months
4: PDR	New or recurring	Referral to tertiary center	Referral to tertiary center
	Stable (after treatment)	6–12 *** months	3–12 months
CSME	New or recurring	Referral to tertiary center	Referral to tertiary center
	Stable (after treatment)	3 *** months	3 months

* Diabetes in control is defined as HbA1c ≤ 53 mmol/mol (7.0%) and blood pressure < 130/80 mmHg. ** If at the first visit, an interval of 24 months is recommended. *** A longer interval is acceptable at the clinician’s discretion. CSME is defined as DME with at least one of the following criteria: (1) macular edema ≤ 500 µm from foveola, (2) hard exudates ≤ 500 µm from foveola, (3) macular edema > 1 disc area with any part of the edema extending within 1 disc diameter from foveola. Abbreviations: CSME = clinically significant macular edema; DME = diabetic macular edema; DR = diabetic retinopathy; PDR = proliferative diabetic retinopathy; NPDR = nonproliferative diabetic retinopathy.

**Table 2 jpm-13-00546-t002:** Age- and sex-stratified incidence rates of patients with diabetic macular edema commenced in anti-VEGF therapy during 2012–2021. All numbers are stated per 100.000 inhabitants within the given strata.

Age Category	Males	Females
<40 years	1.2	1.0
40–44 years	7.0	2.4
45–49 years	12.3	5.1
50–54 years	22.4	8.3
55–59 years	27.6	18.5
60–64 years	39.0	23.8
65–69 years	45.2	27.1
70–74 years	52.9	22.1
75–79 years	44.9	27.5
80–84 years	25.2	17.9
85–89 years	24.7	16.3
≥90 years	3.0	10.3

## Data Availability

Data regarding census are publicly accessible and can be found here: https://www.dst.dk/en/, accessed on 18 March 2023. Restrictions apply to the availability of data regarding patients. Data regarding patients were obtained from the BOB database and are available from the authors with the permission of the department head and relevant Danish authorities.

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
