# Peer review of "Anti-VEGF Treatment of Diabetic Macular Edema in Denmark: Incidence, Burden of Therapy, and Forecasting Analyses"

_jpm, 2023, doi:10.3390/jpm13030546_

Round 1

Reviewer 1 Report

In this study, the authors summarize the prognosis of anti-VEGF therapy for diabetic macular edema in Denmark. It also predicts what the number of patients will be in the future. It's clear and reasonable, but only time will tell whether the prediction algorithm is correct and how factors such as new therapeutic drugs are involved, as written in the limitation.

Author Response

Reviewer #1 general comments:

In this study, the authors summarize the prognosis of anti-VEGF therapy for diabetic macular edema in Denmark. It also predicts what the number of patients will be in the future. It's clear and reasonable, but only time will tell whether the prediction algorithm is correct and how factors such as new therapeutic drugs are involved, as written in the limitation.

Authors’ response:

Thank you for your time and comments. We agree.

Reviewer 2 Report

This article provides important insights into the use of anti-VEGF therapy for patients with diabetic macular edema. The temporal trends identified in the article can help healthcare providers and policymakers better design and plan anti-VEGF services for patients with DME. The findings can help healthcare providers understand the patterns of referral and discontinuation, the burden of treatment, and the number of patients requiring active anti-VEGF therapy. Overall, this article is a valuable contribution to the field of diabetic macular edema treatment and can inform clinical practice and health service planning.

But there are some questions:

1. accoding to your conclusion, " The temporal trends identified in the article can help healthcare providers and policymakers better design and plan anti-VEGF services for patients with DME." So what is exactly the better design and plan? ex: extend the follow intervals after first anti-VEGF therapy or ....?

2. according to your conclusion, " The findings can help healthcare providers understand the patterns of referral and discontinuation" the same, what is the referral and discontinuation patterns resulted from your study, can you clear describe that?

Author Response

Reviewer #2 general comments:

This article provides important insights into the use of anti-VEGF therapy for patients with diabetic macular edema. The temporal trends identified in the article can help healthcare providers and policymakers better design and plan anti-VEGF services for patients with DME. The findings can help healthcare providers understand the patterns of referral and discontinuation, the burden of treatment, and the number of patients requiring active anti-VEGF therapy. Overall, this article is a valuable contribution to the field of diabetic macular edema treatment and can inform clinical practice and health service planning.

Authors’ response:

Thank you for your time and comments.

Reviewer #2 specific comments:

  1. accoding to your conclusion, " The temporal trends identified in the article can help healthcare providers and policymakers better design and plan anti-VEGF services for patients with DME." So what is exactly the better design and plan? ex: extend the follow intervals after first anti-VEGF therapy or ....?

  1. according to your conclusion, " The findings can help healthcare providers understand the patterns of referral and discontinuation" the same, what is the referral and discontinuation patterns resulted from your study, can you clear describe that?

Authors’ response:

Thank you for these comments, which deals with the conclusion of the manuscript, and is better answered in combination. We agree that these sentences need further clarity. Our conclusions are from a more general perspective, i.e. when designing anti-VEGF services, one needs to plan for number of patients and need for therapy of today and for the years to come. The reviewer highlights the details of treatment regimens, which our study have not explored and therefore cannot conclude upon. Our study conclude, that unlike the initial years in which we saw a steady increase in number of patients with DME in active anti-VEGF therapy, we expect that the years to come demand a more stable number of patients with need for anti-VEGF therapy. Therefore, in terms of staffing, resource allocation, injection rooms, etc.; our study expects no significant increase in the years to come, at least until the year 2035. Based on the reviewer comments, we have rephrased and clarified the conclusion of the manuscript.